# OpenReview forum: "SysBench: Can LLMs Follow System Message?"
_ICLR.cc/2025/Conference — ICLR 2025 Poster_

### Official Review · Reviewer_WYwy · 2024-10-23

**Soundness:** 3
**Presentation:** 3
**Contribution:** 3
**Rating:** 6
**Confidence:** 4

**Summary:**

This work presents SysBench - a collection of 500 conversations with different system prompts used to evaluate whether models adhere to constraints laid out in the system prompt. The authors break this dataset into two categories: parallel and dependent. In parallel cases, each turn of the conversation could be modeled separately while in dependent cases later responses depend on earlier turns of conversation. The authors then evaluates a wide range of Language Models on their adherence to the system prompt constraints, using GPT-4o to judge constraint adherence. The aggregate these GPT-4o decisions into 3 metrics based on individual constraint adherence (termed CSR), turn-level complete constraint adherence (termed ISR), and conversation level adherence (termed SSR). Finally, they analyze how adherence varies across different types of constraint, as the conversation progresses, and how attention correlates with adherence to prompt constraints.

**Strengths:**

- Most chatbots are used in multi-turn settings and coherence to system prompts across these seems to be a reasonable extension to work such as IFEval. The authors correctly identify that benchmarking and evaluation in this space seems somewhat lacking v.s. single turn evaluations by comparison.

- If I assume reliability of the LLM-as-a-judge procedure, the investigative analyses in section 4.5 are quite interesting, especially those showing that the distinction between system and user prompt seems to be minimally impactful on results. Understanding at a deeper level what leads to adherence to system prompt instructions seems to be a meaningful basis for developing improvements to adherence.

**Weaknesses:**

Post Discussion Update: Thanks to the authors for their efforts running the cross-model bias comparison and responding to my other points. These responses and additional experiments indeed improve the work. My primary concern is that there still is not statistical significance testing, but overall I think this work is a relatively sound contribution to the space and generally would lean towards an accept decision.

- This benchmark is relatively small size and many of the metrics rely on chains of LLM inference to compute. Combined, these factors seem likely to risk many of the results in the work might not be statistically significant. However, at present the authors don't perform any significance testing on the results.

- The metrics in this work rely entirely on an LLM as a Judge procedure without evaluating or reporting how well correlated the procedure is with human judgement in this context. Furthermore, the work uses only a single closed-source model for this judge which is likely to be biased to specific factors of LLM responses. Without confirmation that these results are either strongly correlated with human judgement or that these results are consistent across LLM Judges from different model families, it doesn't yet seem to have been showed that this evaluation returns a consistent and meaningful metric.

- The description of the data generation process does not explicitly lay out many key details. The source of the original prompts, the mechanisms used to verify data, and the level of synthetic data generation in the creation process is all somewhat vague. I've placed concrete questions in these topics below.

**Questions:**

- How much does the ordering of models depend on the choice of LLM as a Judge and the prompt provided? For example, is the benchmark self correlated if you use any other model (e.g. Claude/Llama/Qwen/Gemini) rather than GPT-4o as the judge?

- In the dependent setting, how do assure that "User" responses are appropriately contextualized to be dependent on what the system responded? If they are also just static responses from the initial data collection, this seems a mismatch to the expectations of the "dependent" setting.

- Section 3.2 " All data are checked independently by multiple experts in multiple rounds to ensure quality." What are the agreement metrics amongst different annotators who checked these independently? How were disagreements resolved? How many annotators checked each example? This will give readers empirical measures of how robust and consistent the decisions made in constructing these datasets were. Reporting an agreement measure such

- In Section 3.2, you reference "from online logs". Where exactly do these come from? Is it from LmSysChat or some other public resource? Since these are the foundation of your dataset, explaining their origin and the data gathering process seems essential.

- Section 3.2 "For each system message, we collect corresponding user conversations and use GPT-4o to assist in generating conversations." This is unclear to me. Are the conversations collected from users or generated by GPT-4o? If they were not generated by GPT-4o, what level of assistance did this model provide? Again, this data is the foundation of all the following results so explaining how and where GPT-4o was used for assistance is critical.

- Section 3.3. " To ensure that LLMs can objectively and accurately assess the constraint satisfaction conditions, we manually annotated evaluation checklists". What was the agreement between the LLM-as-a-judge and the manual annotations? What metric shows that the LLM as a judge procedure is reliable in this setup?

---

> ### Comment · Reviewer_WYwy · 2024-11-27
> **Please Respond!**
>
> Hello Authors! I want to highlight that the concrete questions that I have listed in my review are, in my view, core to understanding and reviewing this work. Many of these questions are seeking clarification the methods used, not additional experiments and as such **can likely be addressed with writing improvements**. My original scores reflected this belief that these would be likely addressable points, but that the variance in LLM-as-a-judge is a perhaps harder to address concern that may undermine the soundness..
>
> However, given the lack of engagement with the present review process, I am concerned that this work, if published, would not make key improvements in presentation to clarify common confusions between both myself and Reviewer 3kVa. As such, I am reducing my presentation and rating scores to reflect this concern!
>
> I want to be clear that, given the extended discussion period, these concerns can likely be clarified via the discussion process but at present they are not.

---

> > ### Author Response · Authors · 2024-11-29
> >
> > >Q1: How much does the ordering of models depend on the choice of LLM as a Judge and the prompt provided? For example, is the benchmark self correlated if you use any other model (e.g. Claude/Llama/Qwen/Gemini) rather than GPT-4o as the judge?
> >
> > In response to your query, we have expanded our evaluation to include a variety of judges beyond GPT-4o. Specifically, we have incorporated GPT-4 Turbo, Claude-3.5-Sonnet, and Qwen2.5-72b-Instruct as evaluators in our study. These additional LLMs were selected to assess the performance of the four models that demonstrated the closest and best performances in our main experiments: GPT-4o, GPT-4 Turbo, Claude-3-Opus, and Qwen2.5-72b-Instruct.
> > The results from these expanded evaluations indicate consistent model rankings across different judges and stable evaluation metrics. This consistency confirms the fairness and reliability of our benchmark evaluation, suggesting that our findings are robust irrespective of the specific LLM judge or the prompts used.
> >
> > | Model                 | GPT4o  evaluator    | GPT4-Turbo evaluator | Claude-3.5 evaluator | Qwen2.5-72B-Instruct evaluator |
> > |-----------------------|------------|------------|------------|----------------------|
> > | **GPT4o**             | **87.1%**  | **87.5%**  | *86.7%*    | **90.2%**            |
> > | **GPT-4-Turbo-20240409** | *86.5%*  | *87.4%*    | **86.9%**  | **90.2%**            |
> > | **Claude-3-Opus**     | 85.0%      | 85.3%      | *86.7%*    | *89.6%*              |
> > | **Qwen2.5-72B-Instruct** | 80.4%    | 81.4%      | 80.6%      | 87.0%                |
> >
> >
> > >Q2: In the dependent setting, how do assure that "User" responses are appropriately contextualized to be dependent on what the system responded? If they are also just static responses from the initial data collection, this seems a mismatch to the expectations of the "dependent" setting.
> >
> > In the construction of our dataset, we considered the influence of the uncertainty in assistant responses on the objectivity of evaluations. To address this, our design of system messages and user queries avoids dependent settings that require specific references to historical assistant responses.
> >
> > For example, in system ID 240, the system message includes settings such as: "You are a senior test engineer interviewer at a top technology company," "After two rounds of interview, ask the candidate if they wish to know more about the company," and "If the candidate has no further questions after two rounds of interview, provide a comprehensive evaluation of the candidate." Thus, regardless of the content of previous assistant responses, after two rounds of interview, if the interviewee (user) says there are no more questions, a comprehensive evaluation based on the historical dialogue is required. Additionally, the third-round user query for system ID 241 is "Can you explain the example you just mentioned?" ensuring that the response inherently relates to the previous round, regardless of its content.
> >
> > This context-dependent, open-ended question setting ensures that current round questions must integrate historical dialogue, reflecting the dependent setting. For more detailed information about our dataset, please refer to our supplementary file .datas/system_benchmark_eval_datas.json
> >
> > >Q3: Section 3.2 " All data are checked independently by multiple experts in multiple rounds to ensure quality." What are the agreement metrics amongst different annotators who checked these independently? How were disagreements resolved? How many annotators checked each example? This will give readers empirical measures of how robust and consistent the decisions made in constructing these datasets were. Reporting an agreement measure such
> >
> > Initially, each data entry was annotated by one of 21 trained annotators. To ensure quality and consistency, entries are then reviewed by five quality inspectors, and each piece of data is checked by two inspectors. If any data entry failed to meet the quality standards during these checks, it was sent back to the initial stage for further refinement. To quantify data validation criteria, a final sample of 20 entries (20 system messages and corresponding 100 user conversations) undergoes human verification of evaluation reliability, which achieving a consistency rate of of 97.3% at the constraint granularity and 94% at the instruction granularity. More details can be find in Appendix B of our revised pdf.

---

> ### Author Response · Authors · 2024-11-28
>
> Thank you very much for your attention to our work and for your constructive comments. We sincerely apologize for the delay in submitting our rebuttal, which was due to the time required to coordinate experimental resources to complete the model bias verification. We consider this issue is crucial and wanted to provide empirical evidence to support our findings. To address your concerns, we have thoroughly elaborated on the data construction process and the validation of the evaluation procedure in Appendix B and Appendix C, respectively, of our revised manuscript. These additions have significantly contributed to improving the rigor of our study.
>
> Here are our detailed responses to each of the issues you raised:
>
> >W1: This benchmark is relatively small size and many of the metrics rely on chains of LLM inference to compute. Combined, these factors seem likely to risk many of the results in the work might not be statistically significant. However, at present the authors don't perform any significance testing on the results.
>
> Our dataset includes 500 system messages and 2,500 user conversations. Compared to instruction following benchmarks like IFEval, which usually involve hundreds of user instructions, the size our dataset is considerably large. The system messages in our dataset are rich in contextual information and complex constraints, which require multiple rounds of user queries, thereby adding to the challenges in data construction. Our data's length and complexity are sufficient to reflect real-world challenges, as can be seen in Appendix D's SAMPLE DATA FORMAT. (Please note that the data in Figure 2 workflow is not our actual sample data but a highly simplified version for presentation purposes.)
>
> >W2: The metrics in this work rely entirely on an LLM as a Judge procedure without evaluating or reporting how well correlated the procedure is with human judgement in this context. Furthermore, the work uses only a single closed-source model for this judge which is likely to be biased to specific factors of LLM responses. Without confirmation that these results are either strongly correlated with human judgement or that these results are consistent across LLM Judges from different model families, it doesn't yet seem to have been showed that this evaluation returns a consistent and meaningful metric.
>
> It is important to clarify that our evaluation mechanism does not entirely depend on an LLM. In practice, we have observed that directly inputting results into an LLM for evaluation can indeed lead to unstable outcomes. To address this, we have implemented a rigorous evaluation checklist for each user query during the data construction phase. This checklist is designed to ensure the objectivity and stability of our evaluations. Further details on this process can be found in our response to Question 6.
>
> In response to your valuable suggestion, we have conducted additional experiments to validate model bias and have included a detailed analysis of human consistency validation in Appendix C of our paper. These additions demonstrate that our metrics show high consistency with human judgement and confirm that the results are robust across different LLM judges from various model families.
>
> >W3: The description of the data generation process does not explicitly lay out many key details. The source of the original prompts, the mechanisms used to verify data, and the level of synthetic data generation in the creation process is all somewhat vague. I've placed concrete questions in these topics below.
>
> To address your concerns and provide clarity, we have included Appendix B in the revised manuscript. This appendix comprehensively details the data construction process. It covers the sources from which the original prompts were derived, the methodologies implemented to verify the data, and the specific role that synthetic data generation plays in our process. We hope that the additions in Appendix B will satisfactorily answer your questions and provide the necessary transparency regarding our data generation methodologies.

---

> ### Author Response · Authors · 2024-11-29
>
> >Q4: In Section 3.2, you reference "from online logs". Where exactly do these come from? Is it from LmSysChat or some other public resource? Since these are the foundation of your dataset, explaining their origin and the data gathering process seems essential.
>
> Our original system messages are partly collected from the user query log of our llm website, and the other part is manually constructed by our team's experts. Please refer to appendix B for more detailed information.
>
> >Q5: Section 3.2 "For each system message, we collect corresponding user conversations and use GPT-4o to assist in generating conversations." This is unclear to me. Are the conversations collected from users or generated by GPT-4o? If they were not generated by GPT-4o, what level of assistance did this model provide? Again, this data is the foundation of all the following results so explaining how and where GPT-4o was used for assistance is critical.
>
> For the set of 500 system messages, we utilized two instances of GPT-4o, each playing distinct roles to generate multi-round conversational data. One model assumed the role of a user, tasked with generating questions that challenge the system's following capabilities based on the setting of system message. The other model responded to these queries, thereby simulating a realistic interaction. This process resulted in 10 rounds of dialogue for each system message. Subsequently, annotators refined these conversations, distilling and rewrite them into 5 rounds of high-quality dialogue that effectively test the system's adherence to the specified constraints. The specific prompts used for generating these dialogues are detailed in Appendix B.
>
> It is important to note that while these automatically generated dialogues provided a structured foundation, they were not used as the final evaluation dataset due to the critical need for data quality and objective evaluation. Instead, they served to streamline the manual data construction process, facilitating further refinement and validation.
>
> >Q6: Section 3.3. " To ensure that LLMs can objectively and accurately assess the constraint satisfaction conditions, we manually annotated evaluation checklists". What was the agreement between the LLM-as-a-judge and the manual annotations? What metric shows that the LLM as a judge procedure is reliable in this setup?
>
> The checklist is a binary evaluation protocol annotated for each user query during the data construction process, formatted as [constraint content that needs to be assessed]+[constraint type], as seen in Appendix D SAMPLE DATA FORMAT under @evaluation checklist@. During evaluation, the checklist is used as contextual information input into the LLM evaluator, which must output true/false judgments for all checklist items, as outlined in Appendix C.2's prompt template. Essentially, the checklist serves as crucial auxiliary information to enhance the reliability of model evaluations. The reliability of the LLM judge setup has been illustrated in the aforementioned responses.
>
>
> Overall, our dataset is sufficiently large and of high quality, and the evaluation protocol is fair and reliable. If there are any clarifications needed in our response, or if you have any other questions, we welcome further discussions. We will further organize and supplement the content of the paper in the final version of the PDF. We sincerely apologize for the delayed response to your review comments and once again, thank you for your valuable suggestions.

---

> ### Author Response · Authors · 2024-12-01
>
> Dear Reviewer WYwy,
>
>
> We thank again for your contributions to the reviewing process.
> The responses to your concerns and the corresponding paper revision have been posted. Sorry for the late, because it takes quite long time to conduct more experiments.   Please let us know whether we have properly addressed your concerns. We look forward to your reply and welcome any further questions.
>
>
> Best regards,
>
>
> Authors of Paper SysBench

---

### Official Review · Reviewer_3kVa · 2024-11-04

**Soundness:** 2
**Presentation:** 3
**Contribution:** 2
**Rating:** 3
**Confidence:** 4

**Summary:**

The paper creates SysBench, a collection of 500 system prompts collected from the web to test the ability of different LLMs to follow system prompts. An evaluation is conducted on 16 open and proprietary LLMs. Results show bigger / proprietary models show better performance. Additionally, the paper explains the cause of the failure, by the lack of attention given to the system prompt.

**Strengths:**

1. system messages are of increasing importance, especially in applications. Chat applications from big techs (Chatgpt, Claude, etc.) come with very long system messages to make them function in desired behaviors. The paper proposes a benchmark to evaluate such capability which may direct the interest of the research community towards system message following LLMs.

2. the paper makes effort to explain the performance gap between language models by analyzing the attention layer of different models.

**Weaknesses:**

1. In section 3.2, the paper mentions, “We initially collect thousands of system messages from online logs, and filter out duplicate and noisy data based on heuristic rules and clustering.” more explanation should be provided on the dataset construction process. Examples of " annotation guidelines designed by experts" and demographics of the 21 trained data annotators may be provided in the appendix to support this.

2. In section 3.2, the paper mentions that each system message is related to 2-3 system constraints, hence presenting a moderate level of complexity.  However, compared to page-long system prompts often used from big techs, 2-3 system constraints look a bit short.

3. The paper lacks a detailed analysis on the performance of different models. Figure 3 breaks down the dataset into different domains, but this does not seem to be reflected in the evaluation.

4. Another widely used application for system messages is safety. Chat applications often want the model to refuse to respond to some user instructions. The benchmark could have been better by including such examples.

5. Analysis between attention distribution and performance is very interesting. However, the paper only presents if for three models. It would have been insightful if the results for other models were also included in the appendix.

6. An error analysis on when models fail can be added to provide more insight on improving models in system message following.

**Questions:**

see weaknesses

---

> ### Author Response · Authors · 2024-11-28
>
> Thank you very much for your thoughtful feedback and constructive questions regarding our manuscript. We truly appreciate the time and effort you have invested in reviewing our work. Based on your valuable input, we have updated the revised PDF and included additional details to address your concerns. Below, I am pleased to provide a detailed response to each of your questions:
>
> **Q1. Process of Data Construction**:
> >In section 3.2, the paper mentions, “We initially collect thousands of system messages from online logs, and filter out duplicate and noisy data based on heuristic rules and clustering.” more explanation should be provided on the dataset construction process. Examples of " annotation guidelines designed by experts" and demographics of the 21 trained data annotators may be provided in the appendix to support this.
>
> Our original system messages were gathered from real-world user query logs and manually constructed by experts on our team, who possess extensive experience in large model agent development. The data construction process involved 21 annotators, 5 quality inspectors and 3 experienced data science experts and took 18 days. The comprehensive data construction process and more details (such as annotation guidelines) can be found in Appendix B. In addition, we demonstrated the high quality of our data and the reliability of our evaluation results through experiments in Appendix C.
>
> **Q2. Data Complexity**:
> >In section 3.2, the paper mentions that each system message is related to 2-3 system constraints, hence presenting a moderate level of complexity. However, compared to page-long system prompts often used from big techs, 2-3 system constraints look a bit short.
>
> There might have been some ambiguity in our description, and we appreciate the opportunity to clarify. In Section 3.2, we indicated that each *user instruction* is associated with 2-3 system constraints, *not that the system message* is limited to 2-3 constraints.
>
> To clarify, as demonstrated in Appendix D's SAMPLE DATA FORMAT, many system messages contain as many as six constraints, along with detailed task scenario descriptions. In fact, most of our system messages are comparable in length and complexity to page-long prompts, incorporating rich contextual information and typically featuring six or more constraints. These characteristics are representative of real-world scenarios.
>
> For a more comprehensive understanding, we encourage you to refer to our supplementary file (`.datas/system_benchmark_eval_datas.json`), which provides an extensive view of our dataset, including the structure and complexity of the constraints.
>
> **Q3. Domain-specific Analysis**
> >The paper lacks a detailed analysis on the performance of different models. Figure 3 breaks down the dataset into different domains, but this does not seem to be reflected in the evaluation.
>
> Our experiments provide a comprehensive and in-depth analysis of models' general system message following abilities, focusing on constraint types, instruction alignment, and multiturn stability. While domain-specific performance is indeed a valuable area of study, our work represents the first effort to evaluate and analyze LLMs' ability to follow system messages systematically，which primary focus is intended to address overarching challenges rather than delve into domain-specific performance.
>
> The domain breakdown in Figure 3 is included to highlight the extensive coverage of diverse scenarios in our dataset. It is not, however, the primary focus of our evaluation. We hope that this foundational work will inspire and enable future research to explore system message following in specific domains.

---

> ### Author Response · Authors · 2024-11-28
>
> **Q4. Safety Examples**:
> >Another widely used application for system messages is safety. Chat applications often want the model to refuse to respond to some user instructions. The benchmark could have been better by including such examples.
>
> Thank you for highlighting this important point. In our work, safety-related scenarios are incorporated under the category of **‘misalignment’**, as detailed in the *User Instruction* paragraph of Section 3.1. These scenarios are specifically illustrated in the third round of dialogue within the data sample provided in Appendix D. Additionally, we have conducted experimental analyses on model performance under misalignment conditions in Section 4.3.
>
> We appreciate your suggestion and will consider further emphasizing the scenarios in our paper to make this aspect more explicit.
>
>  **Q5. Attention Distribution Analysis**:
>  >Analysis between attention distribution and performance is very interesting. However, the paper only presents if for three models. It would have been insightful if the results for other models were also included in the appendix.
>
> Thank you for your appreciation and interest in our experimental exploration. Due to the closed-source nature of some models, our attention score experiments could only be conducted on open-source models. The open-source models evaluated in our main experiments include the llama series, qwen series, glm4-9b-chat, and mixtral series. We have conducted experiments and analyses on llama-3.1-8b, qwen2-72b, and glm4-9b-chat, covering models in both English and Chinese, and large and small scales, observing phenomena with certain universality. We had planned to include experiments with the mistral model, but due to time constraints, we have not yet obtained results. If you are interested, we can include these additional experiments in the final version of the PDF. We indeed believe this is an interesting issue worth discussing further, and we thank you once again for your recognition.
>
> **Q6. Error Analysis**:
> >An error analysis on when models fail can be added to provide more insight on improving models in system message following.
>
> We agree that understanding when and why models fail can provide valuable insights for improvement. In our study, we observed that models typically struggle with system message following when dealing with constraints that contradict common sense, misaligned instructions, or the constraints are heavily dependent on the historical context of the conversation.
>
> While we understand the value of such an analysis, we also considered the "black box" nature of large models, which may limit the utility of specific case studies in providing actionable insights for model improvement. Besides, the extensive length of our data texts, which include page-long system message coupled with multi-turn long conversations, makes detailed case study presentations impractical. Instead, we have opted to include a set of general guidelines in the discussion section of our paper (Appendix E). These guidelines are intended to provide broader insights on improving models in system message following.
>
>
> **A Summary of Our Responses**:
>
>  Overall, our work has provided a high-quality dataset for evaluating the LLM's capability of following system messages, and it represents the first systematic and comprehensive analysis in this area. We hope our responses alleviate any concerns you may have regarding data quality. We will supplement our work with further attention exploratory experiments and analyses as per your interest. We look forward to your feedback and suggestions.

---

> ### Author Response · Authors · 2024-12-01
>
> Dear Reviewer 3kVa,
>
>
> We thank again for your contributions to the reviewing process.
> The responses to your concerns and the corresponding paper revision have been posted. Sorry for the late, because it takes quite long time to conduct more experiments.   Please let us know whether we have properly addressed your concerns. We look forward to your reply and welcome any further questions.
>
>
> Best regards,
>
>
> Authors of Paper SysBench

---

### Official Review · Reviewer_AWyd · 2024-11-05

**Soundness:** 4
**Presentation:** 3
**Contribution:** 3
**Rating:** 6
**Confidence:** 4

**Summary:**

The paper presents a benchmark  for evaluating how well LLMs follow `system messages.` The authors study three dimensions: 1) constraint violation, 2) instruction mis-judgement/ill-following, and multi-turn (in)stability. The work includes a dataset of 500 system messages with 5-round conversations each, and an evaluation protocol. The authors study 16 LLMs and provide a detailed analysis for the performance.

**Strengths:**

- (to-date) First systematic benchmark for system message following capabilities that appears in the submission. The research seems well-motivated by real-world applications and security concerns.
- Good task type and domain coverage (Figure 3, Table 1), and three-level granularity metrics (CSR, ISR, SSR) for evaluation. The dataset seems to have a balanced distribution.
- Offers an overview of performance comparison across models on different types of sys message constraint types. (Figure 4).

**Weaknesses:**

- While the paper throughly evaluates how well LLMs follow system messages and identifies various issues (constraint violations, instruction misjudgment, and multi-turn instability), it doesn't translate these findings into (potential) practical guidelines for practitioners. If for (Figure 4), that the Style constraints tend to be worse followed than other types; or that the performance degrades over multiple turns, I believe the audience usually expect some attempts to provide mitigation strategies or recommendations.

- Missing justification or citations for line 194--195, and the reliance on GPT-4o as the sole verifier:
> For instance, GPT-4o (OpenAI, 2024) verifier achieves over 94% consistency with human evaluations.
I wonder is this statistics based on your own human study? I found this sentence unclear.

Generally, reliance on GPT-4o as verifier raises concerns, where works identified biases [1] [2]. Is there cases where the verifier may fail? Inclusion of strong open-source models like Llama3.1-405B could be a major inclusion. Human verification could also be valuable.


References:

[1] [OffsetBias: Leveraging Debiased Data for Tuning Evaluators](https://arxiv.org/pdf/2407.06551)
[2] [Large Language Models are not Fair Evaluators](https://arxiv.org/pdf/2305.17926)

**Questions:**

- Do you see degradation in multi-turn conversations is due to the accumulation of errors or genuine forgetting of system constraints?
- Given your findings about instruction alignment, what recommendations would you make for designing system messages that are more robust against misaligned instructions?

---

> ### Author Response · Authors · 2024-11-28
>
> Thank you very much for your insightful comments and suggestions regarding our work. We have thoroughly considered your feedback and have updated the revised PDF accordingly. In response to your valuable suggestions, we have expanded the discussion section in Appendix E to provide guidance on improving system message following ability. Additionally, we have included experiments and details on evaluation protocol in Appendix C, demonstrating the stability and reliability of our evaluation mechanism. Below, we address each of your points in detail:
>
> 1. **Potential Practical Guidelines**
>
> In Appendix E "Discussion and Future Work," we provide generalized guidelines from three perspectives: High-quality Training Data, Rule-based Alignment, and Post-hoc Steering. Specifically, to address the worse following of style constraints, incorporating more high-quality data of this constraint type for fine-tuning and alignment can be a practical method. Enhancing multi-turn stability is a highly challenging issue.We believe that initially, it is essential to improve the general capabilities of the model to reduce the adverse effects of incorrect responses on subsequent dialogues (as shown in Figure 5 of the paper), thus preventing error accumulation. Secondly, dynamically adjusting the allocation of attention during successive turns is a potential approach, which has been studied in [1]. However, it has been observed that allocating more attention to system messages can reduce the quality of responses to user queries, and finding the right balance in practical applications remains a challenge. Overall, following system messages is a highly challenging task in large language models, and there is substantial room for further exploration of principles and solutions. Our primary contribution lies in providing a high-quality evaluation dataset for the first time and systematically conducting a comprehensive analysis of this issue. These suggestions are preliminary, and we hope our dataset will facilitate further in-depth research into these challenges.
>
> 2. **Fairness of Evaluation Procedure**
>
> Following your valuable suggestion, we have added experiments on model bias and detailed human verification in Appendix C.1. The results, which are also addressed in the response to Reviewer WYwy, demonstrate that our evaluation mechanism is stable and reliable, with minimal dependency on the model.
>
> 3. **Degradation in Multi-turn Conversations**
>
> Indeed, theoretically, the degradation in multi-turn performance is a cumulative effect of error accumulation and multi-turn decay. Our multi-turn experiment provides a strict and macroscopic measurement, reflecting the model's overall performance in consistently following system messages. For a deeper exploration of the phenomenon of system message decay, please refer to [1] .
>
> 4. **Designing More Robust System Messages**
>
> In our experience, system messages with detailed task description, clear format, and explicitly stated safety rules can be more robust against misaligned instructions. And we have listed related studies that explore this issue in more depth in Appendix E.
>
> We sincerely appreciate your thorough review and hope that our responses and updates adequately address your concerns. Should you have any further questions or require additional clarifications, please do not hesitate to contact us. We look forward to your continued guidance and feedback.
>
> [ [1] Measuring and Controlling Instruction (In)Stability
> in Language Model Dialogs](https://arxiv.org/pdf/2402.10962)

---

> > ### Comment · Reviewer_AWyd · 2024-12-01
> > **Response to authors**
> >
> > I thank the authors for clarification. I have updated my auxiliary ratings (Soundness 3 --> 4) and my calibration (Confidence 3-->4)

---

> > > ### Author Response · Authors · 2024-12-01
> > >
> > > Dear Reviewer AWyd,
> > >
> > >
> > > Thank you for acknowledging our clarifications and updating your ratings. We appreciate your thoughtful feedback and are glad our responses addressed your concerns. We remain open to further questions or feedback to enhance our research. Thank you once again for your support.
> > >
> > >
> > > Best regards,
> > >
> > > Authors of Paper SysBench

---

### Meta-Review · Area_Chair_xcZK · 2024-12-20

**Metareview:**

This paper presents a novel benchmark, SysBench, for evaluating LLMs' adherence to system messages. The work is the first of its kind and fills a notable gap, warranting publication. However, the paper could benefit from minor revisions: enhancing clarity on data construction, including practical guidelines for practitioners, verifying evaluation metrics against human judgment, and addressing potential verification biases. Although there are some concerns, the paper's contributions to NLP are valuable, and it should be accepted with suggestions for enhancement (see below).

While the paper provides a thorough evaluation, there are areas for improvement. Firstly, the lack of practical recommendations from findings could be addressed by including mitigation strategies or guidelines for system message construction. Secondly, reliance on GPT-4o as the sole verifier raises concerns about bias; a comparison with human judgment should be included. Additionally, explanations of dataset construction, evaluation processes, and attention distribution analysis should be expanded. Incorporating other strong open-source models into the verification process and conducting an error analysis could also enhance the paper's robustness.

**Additional Comments On Reviewer Discussion:**

The reviewers had differing opinions. Reviewer AWyd saw the paper as above the acceptance threshold and appreciated the benchmark's novelty and thoroughness. Reviewer WYwy and Reviewer 3kVa had significant concerns about the reliability of using a single model as a verifier and the clarity of the dataset construction process. The authors provided additional explanations, expanding on the construction process, and reported experiments on human consistency validation. Despite this, there remain questions about statistical significance testing and the need for more practical guidelines. The reviewers' discussions concluded with a lean towards acceptance, but with identified room for improvement.

---

### Decision · Program_Chairs · 2025-01-22

Accept (Poster)